Microbiology **Spectrum**
🔓 | **Open Peer Review** | Human Microbiome | Research Article

# Metagenomic profiling and predictive modeling of the gut microbiome reveal signatures of gestational disease

Genevieve A. Mortensen,[1] Haley Schmidt,[2] Predrag Radivojac,[3] Yuzhen Ye,[1] David M. Haas[2]

**ABSTRACT** The gut microbiome plays a vital role in maternal health and pregnancy outcomes, yet its impact on conditions like gestational hypertension (GH) and gestational diabetes mellitus (GDM) remains poorly understood. This study explores how the gut microbiome differs between pregnant women with these conditions and healthy controls, using metagenomic sequencing to analyze microbial composition and function. Our findings reveal that women with GH and GDM exhibit greater microbiome variability and distinct shifts in bacterial communities compared to healthy pregnancies. Key beneficial bacteria, such as *Bacteroides fragilis* and *Roseburia intestinalis*, were reduced in cases, suggesting potential disruptions in gut-related metabolic and immune functions. In addition to multiple differentially abundant species of Sphingobacterium in cases versus controls, functional analysis indicated changes in carbohydrate and lipid metabolism, reinforcing the microbiome's connection to metabolic health. Furthermore, machine learning models demonstrated promising results in predicting disease status based on microbiome data, underscoring the potential for gut bacteria as potential predictive biomarkers for pregnancy-related conditions. These insights highlight the gut microbiome's role in pregnancy health and suggest it may be a promising target for future interventions aimed at reducing complications and improving maternal-fetal outcomes.

**IMPORTANCE** Gut microbial dysbiosis has been implicated in pregnancy complications, yet most studies rely on 16S rRNA sequencing, which limits resolution and functional insight. Here, using shotgun metagenomic sequencing and machine learning, we identified robust microbial taxonomic and functional signatures that distinguish gestational hypertension and gestational diabetes from healthy pregnancies. A combined feature set enabled accurate classification of disease status, with overlapping features between statistical and predictive frameworks underscoring biological relevance. Altogether, our study defines high-resolution microbiome signatures with translational potential as predictive biomarkers for maternal health, while also providing an open, reproducible analysis pipeline to support future investigations.

**KEYWORDS** gut microbiome, shotgun metagenomics, gestational hypertension, gestational diabetes, machine learning, predictive modeling, maternal-fetal health

T he human gut microbiome plays a critical role in maintaining host health and has been increasingly recognized as a key player in the pathogenesis of diverse chronic and inflammatory diseases (1, 2). Advances in metagenomics have provided unprecedented insights into the composition and functional potential of microbial communities, enabling researchers to unravel complex interactions between the microbiome and host physiology across health and disease states (3, 4).

**Peer Reviewers** Lianmin Chen, University of Groningen, Groningen, Netherlands; Hongju Chen, Chinese Academy of Sciences, KunMing, China; Jian Zhou, Chinese Academy of Sciences, Beijing, China

Address correspondence to David M. Haas, dahaas@iu.edu.

The authors declare no conflict of interest.

Pregnancy represents a particularly unique physiological state, marked by profound immunological, hormonal, and metabolic shifts that can dynamically alter the maternal gut microbiome (5, 6).

These changes are believed to support fetal development and immune tolerance but may also create vulnerabilities that predispose some individuals to pregnancy-related complications.

Among these complications, gestational hypertension (GH) and gestational diabetes mellitus (GDM) are two of the most prevalent and clinically significant disorders. Both conditions are associated with increased risks of maternal cardiovascular disease, preterm birth, fetal overgrowth, and long-term metabolic disturbances in offspring (7). Despite their prevalence, early diagnostic markers for GH and GDM remain limited, and their pathophysiology is not fully understood. Increasing evidence points to a central role of the gut microbiome in modulating metabolic and inflammatory pathways involved in these diseases (8–10).

Studies have reported alterations in gut microbial composition and reduced microbial diversity in pregnancies complicated by GH and GDM (6, 11). Several pro-inflammatory and metabolically disruptive taxa—along with associated changes in short-chain fatty acid (SCFA) production, bile acid metabolism, and glucose regulation—have been implicated in the onset and progression of these conditions (12). However, many of these findings rely on 16S rRNA gene sequencing, which lacks the resolution and functional insight provided by shotgun metagenomic sequencing. Moreover, there is a paucity of well-controlled studies that directly compare both taxonomic and functional features or their overlap across GH, GDM, and healthy pregnancies using high-resolution metagenomic data (13). Although computational methods, including PICRUSt2 and Tax4Fun2, have been developed to infer functional profiles from 16S rRNA-based sequencing data, a benchmark study showed that 16S rRNA gene-based functional inference tools lack the sensitivity to delineate health-related functional changes in the microbiome (14–16).

To push beyond univariate associations, some microbiome studies have adopted machine learning (ML) approaches to extract multivariate microbial signatures, perform classification tasks, and integrate multi-omics data sets (17, 18). ML models are particularly well-suited to microbiome research because they can capture nonlinear relationships and interactions among thousands of microbial and metabolic features while handling compositional and sparse data structures. These methods have been successfully applied in diverse disease contexts, including metabolic and inflammatory disorders, to improve diagnostic accuracy and identify microbial biomarkers of disease progression (17–19).

One illustrative example is a pooled analysis of metagenomic data across multiple cohorts to identify microbial signatures of periodontitis using a decision tree classifier trained on taxa highly associated with disease (20). Their interpretable ML framework achieved excellent classification performance, demonstrating the power of combining high-resolution sequencing data with transparent predictive models. Similar principles have been applied to gestational disease research, for instance, random forest models on metagenomic linkage groups to distinguish GDM from control pregnancies, revealing disease-specific microbial and inflammatory signatures (21). Recent review literature also emphasizes the growing integration of ML into gestational diabetes care, highlighting its potential in early screening, diagnosis, and risk stratification (22).

In this study, we leverage shotgun metagenomic sequencing to perform a comprehensive, multi-level analysis of the maternal gut microbiome in a cohort of pregnant women, including those diagnosed with GH, GDM, and healthy controls. By analyzing fecal samples collected during the second and third trimesters, we aim to (i) identify disease-specific taxonomic shifts, (ii) evaluate differential microbial functional potential, and (iii) assess the predictive utility of microbiome features in distinguishing between healthy and diseased states.

This work builds upon emerging efforts to establish the microbiome as a non-invasive biomarker source in maternal–fetal medicine. By incorporating interpretable

machine learning approaches and pathway-level functional analysis, we seek to enhance our understanding of the microbiome's role in gestational disease development and progression. Our findings may inform future strategies for early detection, personalized risk stratification, and microbiome-targeted interventions, ultimately contributing to improved health outcomes for both mothers and their children (12, 23).

## MATERIALS AND METHODS

### Study design and data description

This study is a planned analysis of a prospective observational cohort, the Hoosier Moms Cohort (HMC). Full details of the cohort are reported elsewhere (23). Patients receiving prenatal care from outpatient clinics of seven local health organizations were recruited through self-referral, provider referral, and research staff member prescreening of scheduled visits. To ensure statistically significant levels of participants diagnosed with GDM, recruitment efforts for the HMC were prioritized to include potential participants of high-risk populations (e.g., history of GDM in prior pregnancy). Individuals enrolled in the HMC were able to provide informed consent in English or Spanish, at least 18 years of age, and pregnant with a singleton gestation less than 20 weeks of gestational age confirmed by the American College of Obstetrics and Gynecology (ACOG) ultrasound dating guidelines. Patients with a pre-gestational diagnosis of type 1 or type 2 diabetes, a screening HbA1c ≥6.5%, or two abnormal values on a 3-hour oral glucose tolerance test before 20 weeks of gestation were excluded. Other exclusions included pre-pregnancy (1 month prior to conception) chronic (>2 weeks) systemic steroid use, planned pregnancy termination, or the presence of major fetal anomalies discovered prior to enrollment.

Self-collected stool samples were obtained from participants. Samples could be collected at home and brought to a study team member at a clinic or study visit, or participants were provided with a mail-in prepaid kit. Stool samples from pregnant participants were collected during their first and third trimesters, as well as postpartum. The majority of participants submitted one sample during their first trimester. Participant inclusion occurred in two batches. Batch 1 included 50 participants, while batch 2 included 23 participants. A total of 42 case samples and 31 control samples were analyzed. Cases were defined as any participant diagnosed with either GDM or GH. Among the case samples, 35 were from patients diagnosed with GH, while the remaining 18 were from patients presenting with GDM. Some participants were diagnosed with both conditions. Participants' ages ranged from 18 to 39 years. Patient race was self-reported as White, Black, or Other. Participant body measurements were taken to determine body mass index (BMI) at each study visit. Tobacco use was collected at each visit. GH and GDM often co-occur, and among the 18 women with GDM, 11 also had GH. Given the partial diagnostic overlap between GH and GDM and the limited subgroup sample sizes, we performed the primary analyses using a combined case group. In addition, stratified analyses comparing controls to GH and GDM separately were conducted as sensitivity analyses and are provided in Fig. S7 through S12. These analyses revealed shared microbial markers between the GH and GDM groups (see Results), further supporting our rationale for the combined analysis.

### Sample processing and sequencing

Fecal samples for short-read whole-genome DNA sequences were received, processed, and frozen at −80℃, and then sent to the lab when ready for analysis through the Indiana Biobank. Samples were processed at Microbiome Insights using a standardized shotgun metagenomic sequencing workflow. DNA extraction was performed using the Qiagen MagAttract PowerSoil DNA KF Kit with the Thermo Fisher KingFisher automatic extractor. Library preparation was conducted using the Illumina Nextera XT DNA Library Prep Kit, targeting 15 million paired-end reads per sample. Full shotgun sequencing was then performed on the Illumina NextSeq 500 platform (24). Quality control details of the metagenomic sequencing data are presented in Table S2.

## Bioinformatics workflow

We developed a modular and reproducible bioinformatics pipeline, MGPipe (https://github.com/ginnymortensen/MGPipe), that unifies widely adopted tools for processing short-read, paired-end metagenomic sequencing data. This pipeline standardizes preprocessing and profiling steps commonly required in microbiome analysis, allowing researchers to streamline their workflows with minimal configuration. Specifically, the preprocessing of short-read whole-genome shotgun (srWGS) DNA sequences included adapter trimming and quality filtering using fastp (25), followed by the removal of human host DNA using Bowtie2 (26). For taxonomic profiling, we applied Kraken2 (27) for read classification and used Bracken (28) to estimate species-level abundances. Functional profiling of the microbial communities was conducted using HUMAnN3 (29), which characterizes gene family and pathway-level contributions. The complete pipeline was implemented as a single shell wrapper script that automates each stage, manages R and Python dependencies via Conda, and maintains a unified logging structure for reproducibility. In this work, we applied MGPipe uniformly to all samples, including both sequencing batches by differential platform handling, ensuring standardized processing throughout.

## Statistical analysis

To assess and correct for batch effects in the taxonomic abundance data due to differing Illumina platform sequencing between batches, we conducted a two-stage analysis combining permutational multivariate analysis of variance (PERMANOVA) with linear modeling. First, Bray–Curtis dissimilarity matrices were computed from raw taxonomic count data to quantify pairwise ecological distances between samples. PERMANOVA was then performed using batch as the sole explanatory variable (30). This initial analysis revealed a significant batch effect ($R^2 = 0.151$, $P = 0.001$).

To correct for this confounding technical effect, we modeled abundance data using a linear design that included both batch and biological group (case vs. control) as covariates. Raw count data were normalized using the trimmed mean of M-values (TMM) method (31). We then computed $\log_2$-transformed counts per million (logCPM) and associated observational-level precision weights using voom (32), enabling the use of linear models. Batch-associated variation was removed by regressing out the fitted contribution of batch from the transformed abundance matrix. The corrected data were then exponentiated to return values to the original scale.

To confirm the effectiveness of this correction, PERMANOVA was repeated using the adjusted abundance matrix. After correction, the batch effect was no longer statistically significant ($R^2 = 0.016$, $P = 0.273$), and the variance explained by batch dropped from 15.1% to 1.6%. This result supports the effectiveness of the correction procedure and ensures that downstream analyses reflect true biological differences between groups rather than technical artifacts. No batch effect was observed in the functional pathway abundance data following PERMANOVA testing ($R^2 = 0.003$, $P = 0.937$). Procedure results are visualized in Fig. S1 (See Supplementary material).

## Differential abundance testing

We performed differential abundance testing at both the functional (pathway) and taxonomic (species) levels to identify microbial features associated with case status. We used the Phyloseq package in R to organize taxonomic features into usable data for downstream analyses (33). For taxonomic analysis, raw count data from the microbial phyloseq object were input into the DESeq2 framework, which models count data using a negative binomial distribution to account for overdispersion typically observed in sequencing-based experiments (34). The design formula specified group status (Case vs. Control) as the primary variable of interest. DESeq2 estimates size factors to normalize for differences in sequencing depth between samples, then estimates dispersion parameters for each feature using a shrinkage approach to stabilize variance estimates, particularly for low-abundance taxa.

Differential abundance testing was performed using the Wald test, yielding $\log_2$ fold changes for each feature representing the magnitude and direction of differential abundance between case and control groups. A positive $\log_2$ fold change indicates enrichment in case samples, while a negative value indicates enrichment in control samples. *P*-values from the Wald test were adjusted for multiple comparisons using the Benjamini–Hochberg false discovery rate (FDR) procedure. Features with FDR-adjusted *P*-values (*P*adj) less than 0.05 were considered statistically significant.

For functional profiling, relative abundance tables generated by HUMAnN3 were analyzed separately. Pathways were compared across groups using similar DESeq2-based modeling, with pseudocounts and geometric mean normalization applied to ensure compatibility with the negative binomial model. Pathway-level differential abundance was interpreted using the same $\log_2$ fold change and FDR-adjusted *P*-value framework described above.

## Predictive modeling framework

To assess the predictive value of microbial features, we trained and evaluated a panel of six classifiers—XGBoost (XGB), Random Forest (RF), Logistic Regression (LR), Support Vector Machine (SVM), K-Nearest Neighbors (KNN), and Multi-Layer Perceptron (MLP)—on three different input feature sets: taxonomic profiles ($X_{taxa}$), functional pathway profiles ($X_{pathways}$), and their combination ($X_{combined}$)—to create a total of 18 distinct predictive models. Each model was implemented in scikit-learn using standardized supervised learning workflows. Each profile represented feature-level attributes, including relative abundance, false discovery rate, and $\log_2$ fold change, concatenated to capture both quantitative and statistical significance information. Models were trained to predict the binary disease label vector *y*, corresponding to each subject's disease-state status.

Given the small sample size (*n* = 73; 57% cases and 43% controls), we employed a stratified 80/20 train/test split (random seed 42) to preserve class proportions and ensure balanced evaluation. Within the training subset, fivefold cross-validation with grid search was used to tune model-specific hyperparameters, optimizing the Area Under the Receiver Operating Characteristic Curve (AUC) as the performance criterion. The final optimized models were then evaluated on the held-out test set, reporting accuracy, precision, recall, F1-score, and test AUC.

To further assess generalization and model robustness, we conducted Monte Carlo cross-validation by re-sampling 100 stratified train/test splits (seeds 0–99) and applying each previously trained model to new 20% test subsets. For each iteration, ROC curves were computed and interpolated over a common false-positive rate (FPR) grid to derive mean true-positive rates (TPRs) and standard deviation bands, producing smoothed ROC curves with confidence intervals. This resampling approach quantifies how consistently models discriminate under varying data partitions as a critical consideration in small-cohort microbiome studies prone to sampling noise.

## RESULTS

### General cohort characteristics reflect the clinical impact of gestational diseases

Participant characteristics are summarized in Table 1. To evaluate statistical significance between case and control groups, numerical variables were subjected to a Student's t-test, and categorical variables were subjected to a Fisher's exact test. The final data set included 73 fecal samples from pregnant individuals, comprising 31 control participants with healthy pregnancies and 42 participants with gestational conditions classified as cases. Maternal age did not significantly differ between groups (Control: 27.16 ± 10.12 years; Case: 29.43 ± 6.96 years; *P* = 0.288). However, the case group exhibited significantly higher BMI compared to the control group (32.13 ± 12.49 vs. 23.96 ± 10.33; *P* = 0.003), consistent with known risk factors for gestational disease.

**TABLE 1** Cohort characteristics stratified by case status

| Variable | Control ($n = 31$) | Case ($n = 42$) | $P$-value[a] |
|---|---|---|---|
| Age (years) | 27.16 (10.12) | 29.43 (6.96) | 0.288 |
| BMI (kg/m$^2$) | 23.96 (10.33) | 32.13 (12.49) | **0.003** |
| Race (%) | | | 0.935 |
| Black | 7 (22.6%) | 9 (21.4%) | |
| Other | 4 (12.9%) | 4 (9.5%) | |
| White | 20 (64.5%) | 29 (69.0%) | |
| Tobacco use (%) | | | 1.000 |
| No | 21 (67.7%) | 28 (66.7%) | |
| Yes | 10 (32.3%) | 14 (33.3%) | |
| Batch (%) | | | 0.312 |
| Batch 1 | 19 (61.3%) | 31 (73.8%) | |
| Batch 2 | 12 (38.7%) | 11 (26.2%) | |
| Visit (%) | | | 0.816 |
| 1 | 16 (51.6%) | 22 (52.4%) | |
| 2 | 13 (41.9%) | 18 (42.9%) | |
| 3 | 2 (6.5%) | 2 (4.8%) | |
| Delivery EGA (weeks) | 39.17 (0.91) | 38.10 (2.31) | **0.009** |
| Birth weight (g) | 3,380.60 (338.46) | 3,074.74 (703.87) | **0.017** |
| Preeclampsia during labor (%) | 1 (3.2%) | 12 (28.6%) | **0.002** |

[a]Significant $P$-values are bolded.

The distribution of self-reported race was similar across groups, with White individuals comprising the majority in both cohorts (Control: 64.5%; Case: 69.0%), and no significant differences were observed in overall racial composition ($P = 0.935$). Tobacco use was also balanced between groups (Control: 32.3%; Case: 33.3%; $P = 1.000$), as was sample collection batch (Control: 61.3% in Batch 1; Case: 73.8% in Batch 1; $P = 0.312$). The distribution of sample collection visits, used as a proxy for trimester, did not differ significantly ($P = 0.816$).

Notably, clinical outcomes associated with gestational disease showed significant differences. The average estimated gestational age (EGA) at delivery was significantly lower in the case group ($38.10 \pm 2.31$ weeks) compared to the control group ($39.17 \pm 0.91$ weeks; $P = 0.009$), suggesting earlier deliveries among affected individuals. Birth weight was also significantly lower in the case group ($3074.74 \pm 703.87$ grams) compared to controls ($3,380.60 \pm 338.46$ grams; $P = 0.017$), further indicating potential adverse perinatal outcomes in gestational disease cases. Preeclampsia during labor was markedly more common in the case group (28.6%) compared to controls (3.2%; $P = 0.002$).

Collectively, these results support the interpretation that key differences between groups reflect the clinical impact of gestational disease rather than baseline demographic or procedural differences. Additional comparisons, including maternal vital signs and laboratory values, are presented in Table S1.

## Diseased gut microbiome diversity and composition differ from the healthy gut during pregnancy

To assess microbiome diversity and taxonomic structure across gestational disease status, we conducted a comprehensive alpha and beta diversity analysis followed by taxonomic profiling. We aimed to evaluate whether individuals with gestational disease (case group) exhibit distinct microbial community characteristics compared to controls with uncomplicated pregnancies. Stratified analyses revealed largely overlapping microbial signatures in GH and GDM relative to controls. Of the eight statistically significant differentially abundant microbial species in the GDM cohort, five of them are also differentially abundant in the GH cohort with a threshold of $P < 0.05$ (Fig. S7 through S12). This result supports the rationale for the combined case analysis.

We first evaluated community composition using beta diversity metrics. Bray–Curtis dissimilarities were calculated on species-level relative abundance profiles derived from Bracken-processed Kraken2 outputs. Principal Coordinates Analysis (PCoA) and Non-Metric Multidimensional Scaling (NMDS) ordination were performed using the Bray–Curtis distance measure. Ordination plots were overlaid with 95% confidence ellipses to depict group dispersion (Fig. 1A and B).

Statistical comparisons of group separation were conducted using both PERMANOVA and MANOVA. PERMANOVA was applied directly to the Bray–Curtis distance of the microbial profile via adonis2 with 999 permutations with a non-significant result of $P = 0.428$, indicating no major differences between case and control groups. MANOVA was applied directly to the first two principal components of the PCoA and did not reveal significant differences in microbial community composition between case and control groups ($P = 0.254$). Similarly, MANOVA applied to the first two NMDS axes yielded a non-significant result ($P = 0.59$). The NMDS ordination showed an acceptable representation of the data in two dimensions (stress = 0.13), suggesting no gross shifts in beta diversity due to gestational disease status.

We then quantified within-sample alpha diversity using three metrics: Observed richness, Shannon diversity, and Simpson diversity. For species-level profiles, we used native functions via the phyloseq R library to calculate observed richness, Shannon index, and Simpson's index. For pathway-level profiles (from HUMAnN3 unstratified relative abundances), diversity metrics were calculated by summing the number of non-zero pathways after log-transformation. Shannon index and Simpson's index were also calculated natively using the vegan library in R.

Violin plots were generated to visualize group-level distributions of each metric (Fig. 2A and B). Linear models (lm()) and non-parametric Mann–Whitney U tests (wilcox.test()) were used to test for differences in alpha diversity. At the species level, all three diversity metrics were nominally higher in controls compared to cases, but the effect did not remain after adjustment (Observed: $\#P = 4.65 \times 10^{-2}$, $\$P = 9.75 \times 10^{-2}$; Shannon: $\#P = 1.16 \times 10^{-1}$, $\$P = 1.28 \times 10^{-1}$; Simpson: $\#P = 2.46 \times 10^{-1}$, $\$P = 2.62 \times 10^{-1}$). Similarly, pathway-level richness and diversity showed no significant differences following adjustment (Observed: $\#P = 4.67 \times 10^{-1}$, $\$P = 5.58 \times 10^{-1}$; Shannon: $\#P = 5.09 \times 10^{-1}$, $\$P = 2.85 \times 10^{-1}$; Simpson: $\#P = 6.09 \times 10^{-1}$, $\$P = 4.88 \times 10^{-1}$). These

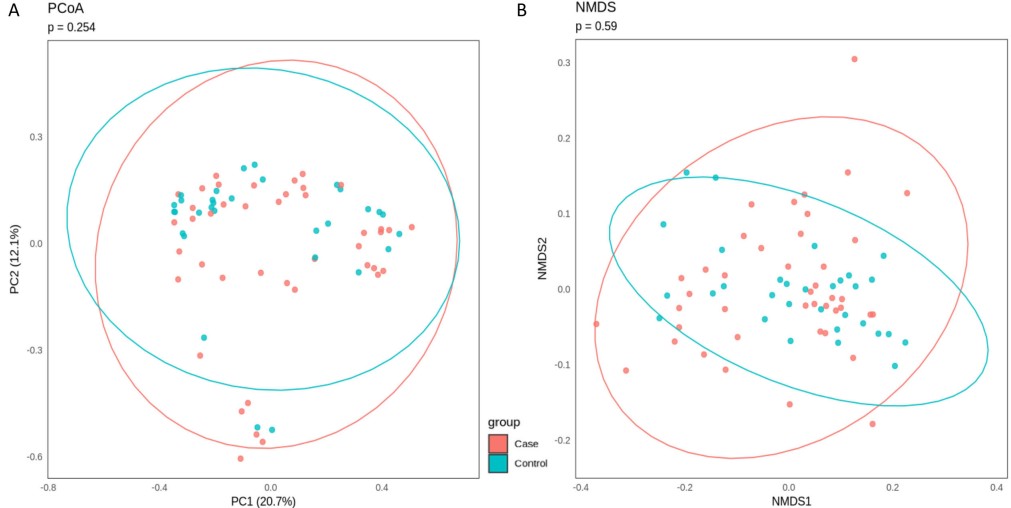

**FIG 1** (A) PCoA plot of Bray–Curtis dissimilarities at the species level. Ellipses show 95% confidence intervals. Significant group separation was not observed ($P = 0.254$). (B) NMDS ordination using the same data, with 95% confidence ellipses. MANOVA applied to the first two NMDS axes yielded no significant group effect ($P = 0.59$, *stress* = 0.13).

results suggest that gestational disease status is not associated with large-scale shifts in within-sample microbial diversity, but may instead reflect more subtle compositional or functional changes, consistent with prior reports of heterogeneous alpha diversity findings alongside compositional alterations in gestational diabetes (35).

A targeted comparison of microbial taxa previously reported to be associated with gestational diseases was performed by evaluating their mean relative abundances across case and control groups (Fig. 3A). The abundance of *Akkermansia muciniphila* was reduced in the case group, consistent with prior studies that report its depletion in patients with GDM (36, 37). In contrast, *Bifidobacterium longum* showed increased abundance in gestational disease cases, which is contrary to reports linking its presence with metabolic health and lower inflammation (38). The genus *Collinsella*, particularly *Collinsella aerofaciens*, was elevated in the case group, supporting previous findings of its association with dysbiosis and GDM-related metabolic perturbations (6). *Faecalibacterium prausnitzii*, a well-known anti-inflammatory commensal, was notably depleted in the case group, in agreement with its previously reported reduction in GDM and preeclampsia (36). Similarly, *Roseburia intestinalis* was reduced in disease cases, aligning with prior studies indicating its loss in inflammatory and hypertensive pregnancy phenotypes (39). Collectively, these findings reflect the loss of protective commensals and over-representation of pro-inflammatory or dysbiosis-associated taxa in gestational disease.

To explore broader taxonomic shifts at the genus level, we plotted the mean relative abundances of the six most dominant genera across case and control groups (Fig. 3B). Overall, genus-level community composition appeared broadly similar between groups; however, subtle group-specific differences in relative abundance were evident. For instance, the genus *Alistipes* was modestly enriched in the control group. This finding is consistent with prior studies reporting an inverse relationship between *Alistipes* abundance and gut inflammation, as well as its reduced prevalence in preeclamptic individuals (37, 40). Conversely, the genus *Segatella* was relatively enriched in the case group, a trend that aligns with prior associations of this taxon with systemic inflammation, impaired glucose tolerance, and metabolic dysfunction (41–43). Direct Case/Control comparisons of selected taxa abundances and *Firmicutes/Bacteroidetes* ratios are presented in Fig. S2 and S3, respectively.

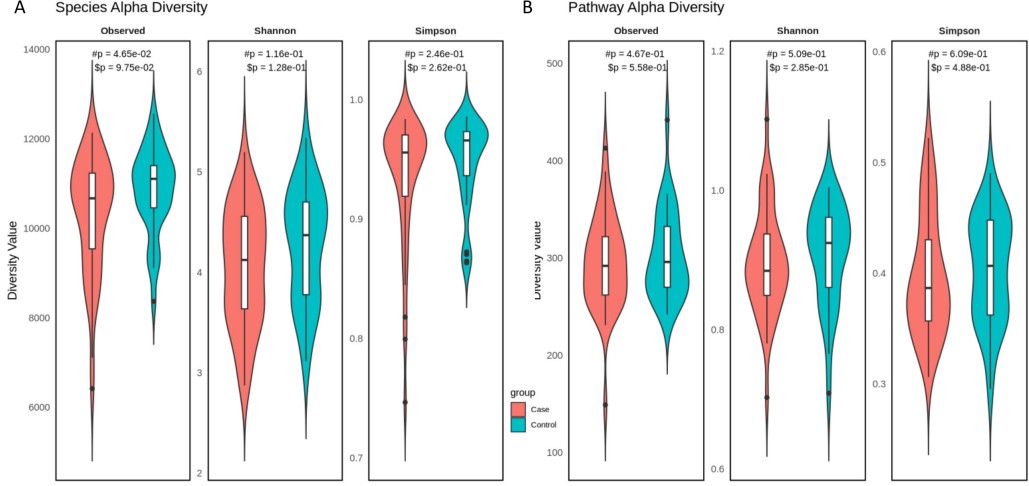

**FIG 2**   (A) Violin plots of species-level alpha diversity metrics (Observed, Shannon, and Simpson) stratified by disease status. All metrics were significantly lower in the case group. *P*-values from linear models (#*P*) and Mann–Whitney tests ($*P*) are shown. (B) Violin plots of pathway-level diversity based on HUMAnN3 output. Functional diversity was reduced in cases across all metrics, but the difference is not statistically significant.

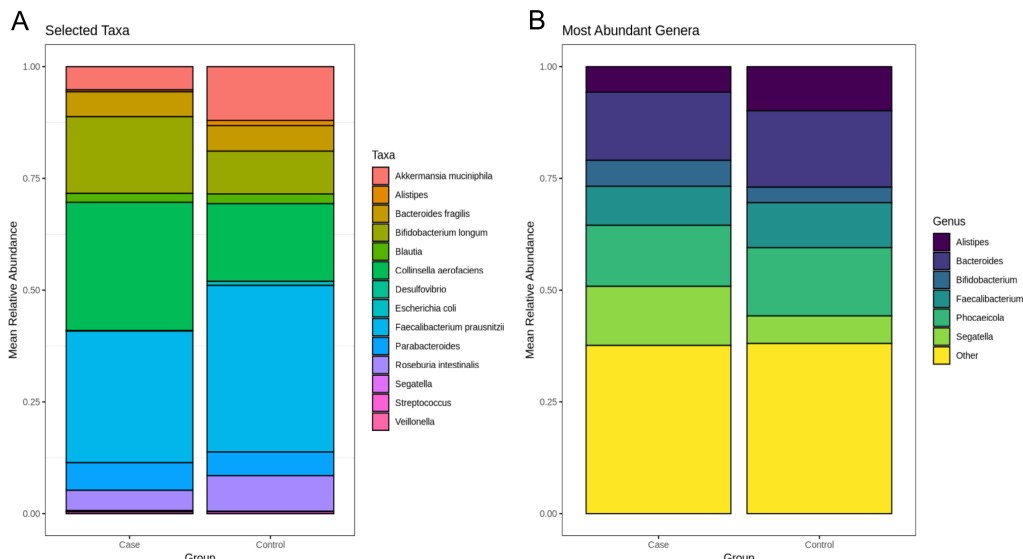

**FIG 3** (A) Stacked bar plot of selected taxa known to be associated with gestational diseases by mean relative abundance. Distinct differences in *Akkermansia municiphila, Bifidobacterium longum, Collinsella aerofaciens, Faecalibacterium prausnitzii,* and *Roseburia intestinalis* abundance proportions are observed between groups. (B) Stacked bar plot of top six most abundant genera by mean relative abundance. Compositional profiles appeared broadly similar across groups, with moderate differences between *Alistipes* and *Segatella*.

## Differentially abundant microbial signatures are associated with gestational disease

Figure 4A displays the top differentially abundant metabolic pathways identified using HUMAnN3. All pathways shown met a significance threshold of FDR < 0.1. Notably, several pathways related to amino acid fermentation and degradation were enriched in case samples, including the *superpathway of L-alanine fermentation* (PROPFERM-PWY), *L-alanine degradation V* (PWY-8189), and *L-alanine degradation VI* (PWY-8188), all of which are linked to the Stickland reaction. Collectively, these results suggest increased alanine-utilizing anaerobic metabolism in Case individuals ($\log_2$ fold changes = 1.28; FDR = 0.049).

In contrast, the *pyruvate fermentation to propanoate II (acrylate pathway)* (PWY-5494) was depleted in Case samples ($\log_2$FC = −0.85, FDR = 0.049), indicating a reduction in short-chain fatty acid production. The *CMP-pseudaminate biosynthesis* pathway (PWY-6143), involved in glycosylation of bacterial flagella, was also less abundant in Case samples, though its FDR did not pass the 0.05 threshold ($\log_2$FC = −1.4, FDR = 0.081). Together, these results suggest that case samples exhibit an altered metabolic profile, with increased reliance on amino acid fermentation and potential decreases in beneficial SCFA production.

Figure 4B illustrates the top 12 differentially abundant microbial species between case and control groups. Among the species significantly enriched in Case samples were *Delmidovirus splanchnicus* ($\log_2$FC = 24.25, FDR = $1.5 \times 10^{-34}$), *Veillonella parvula* ($\log_2$FC = 4.87, FDR = $1.9 \times 10^{-9}$), *Raoultella planticola* ($\log_2$FC = 5.39, FDR = $1 \times 10^{-12}$), and *Ligilactobacillus ruminis* ($\log_2$FC = 4.19, FDR = $1.8 \times 10^{-11}$). Several other species—including *Segatella hominis, Acidaminococcus fermentans*, and *Lacticaseibacillus rhamnosus*—also exhibited significant enrichment in case samples, despite traditionally being associated with commensal or even probiotic roles.

Conversely, only two species were significantly depleted in case samples: *Fusobacterium mortiferum* ($\log_2$FC = −7.59, FDR = $2.6 \times 10^{-24}$) and *Enterobacter hormaechei* ($\log_2$FC = −3.66, FDR = $2.9 \times 10^{-16}$). These taxa showed consistently higher abundances in Control samples across individuals.

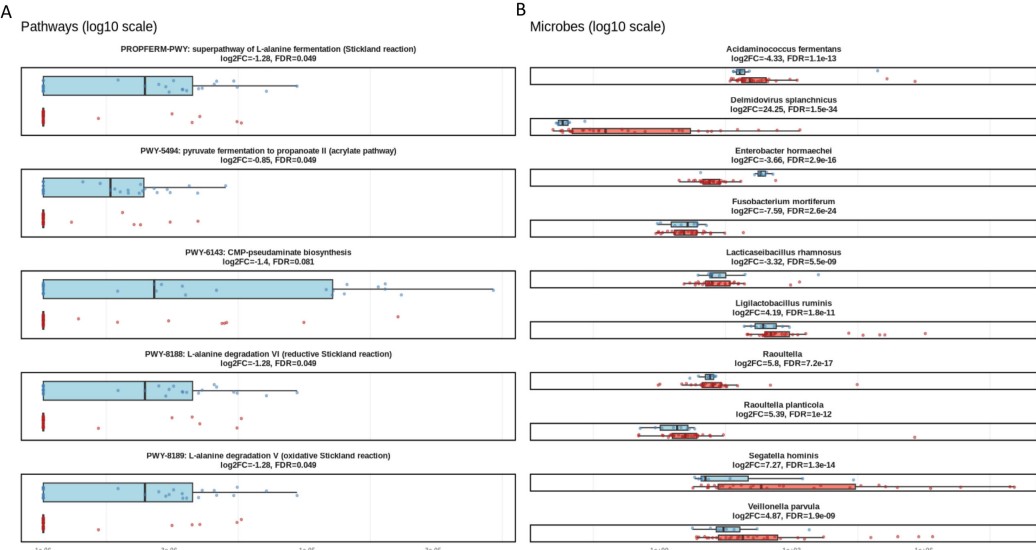

**FIG 4** Each facet displays the $\log_{10}$-transformed relative abundance of a differentially abundant microbial feature. Cases are shown in red and controls are shown in blue. Facet titles indicate the pathway or species name, along with the $\log_2$ fold change (log2FC; Case vs. Control) and the associated FDR. (A) Differentially abundant microbial pathways identified using HUMAnN3. Several pathways related to amino acid fermentation, including those involved in L-alanine degradation and Stickland metabolism, were elevated in case samples. In contrast, short-chain fatty acid biosynthesis and certain nucleotide-related pathways were depleted. (B) Differentially abundant microbial species identified using DESeq2. Case samples exhibited enrichment of taxa including *Veillonella parvula*, *Raoultella planticola*, and *Segatella hominis*, while control samples showed higher abundances of *Fusobacterium mortiferum* and *Enterobacter hormaechei*, indicating compositional shifts in microbial community structure.

Figure 5 displays the significant correlations between significantly differentially abundant pathways and taxa with respect to clinical variables and each other, where all pathways met a significance threshold of FDR < 0.1 and all taxa met an FDR < 0.05 and a $\log_2$FC > 1. Correlations marked with an asterisk ("*") had a $P < 0.05$ and those marked with a plus sign ("+") had a $P < 0.01$, indicating coordinated shifts in both taxonomic and functional profiles in disease. Interestingly, hierarchical clustering of the correlation matrix in Fig. 5A revealed three main functional clusters. The amino acid metabolic pathways (all Stickland reactions) co-clustered and showed strong negative correlations with SCFA producers (e.g., *Lactobacillus gasseri* and *Megasphaera massiliensis*) and SCFA-sensitive genera such as *Klebsiella*.

The other amino acid metabolic pathway, pyruvate fermentation to propionate II (acrylate pathway), appeared to have a dominant and parallel association cluster to the Stickland reaction pathways, showing positive associations with *Akkermansia massiliensis* and *Acetobacter persici*, and clinically significant pathogens (e.g., *Enterobacter hormaechei* and *Fannyhessea vaginae*), in addition to the taxa associated with Stickland reaction pathways. This pattern resembles the dual metabolic restructuring observed in other dysbiotic contexts, in which fermentative energy-yielding pathways coexist with the expansion of mucin-degrading taxa under inflammatory conditions (2).

The enrichment of *Akkermansia* and *Enterobacter* spp. aligns with previous pregnancy-related metagenomic studies showing that mucin-utilizing and opportunistic bacteria increase during gestational metabolic disturbance (6, 9, 10).

In contrast, CMP-pseudaminic acid biosynthesis formed its own branch, positively correlating with *Pseudomonas* spp. and several other pathogenic species, suggesting disease-associated dysbiosis lending to opportunistic pathogen vulnerability. Glycosylation-related pathways such as this have been implicated in bacterial adhesion and immune evasion mechanisms that favor pathogen persistence in inflammatory mucosal environments (2, 4).

The overall pattern suggests a complex restructuring of the microbial community in case subjects, with an overrepresentation of taxa associated with altered amino acid metabolism, SCFA production, and opportunistic growth. The branching between amino acid metabolism versus CMP-pseudaminic biosynthesis suggests a reciprocal effect between microbial populations related to SCFA production and pathogenicity, highlighting the influence of SCFAs on immune cells and the underlying gut vulnerability of diseased patients. Notably, the enrichment of species such as *Veillonella parvula* and *Raoultella planticola*, both previously linked to inflammatory or dysbiotic states, supports the hypothesis of a disease-associated microbial imbalance. These findings echo prior observations that diminished SCFA production and elevated amino acid fermentation is an indicator of metabolic and inflammatory dysbiosis (1, 12).

These findings indicate that case samples exhibit coordinated restructuring of the gut microbiome at both taxonomic and functional levels. To evaluate clinical relevance, we correlated differentially abundant taxa and pathways with host clinical measures in Fig. 5B and C. Both taxa and pathways showed the strongest associations with continuous cardiometabolic biomarkers, particularly lipid-related measures. Several taxa, including *Lysinibacillus* and *Sphingobacterium* species, were positively correlated with total cholesterol, triglycerides, BMI, and systolic blood pressure, forming a shared metabolic signature. Similarly, case-enriched pathways involved in amino acid biosynthesis, phospholipid metabolism, and central carbon metabolism were positively associated with adverse lipid profiles, BMI, and heart rate, while glucose and HDL tended to be inversely correlated. In contrast, categorical pregnancy outcomes and clinical diagnoses showed minimal associations. Together, these results suggest that microbiome alterations in cases are more closely linked to underlying metabolic physiology than to binary disease labels, with additional differential abundance visualizations provided in Fig. S4 and S5.

## Predictive modeling reveals prominent indicators of disease

Using the modeling framework described in Materials and Methods, we next evaluated classifier performance across feature sets and algorithms. Across models, the combined feature set consistently yielded superior performance relative to taxa-only or pathways-only inputs, underscoring the complementary predictive value of taxonomic and functional features (Fig. 6A). The highest overall discriminatory power (test AUC = 0.81) was achieved by the linear SVM trained on combined features, suggesting that a sparse, high-dimensional feature space effectively captured discriminative microbial

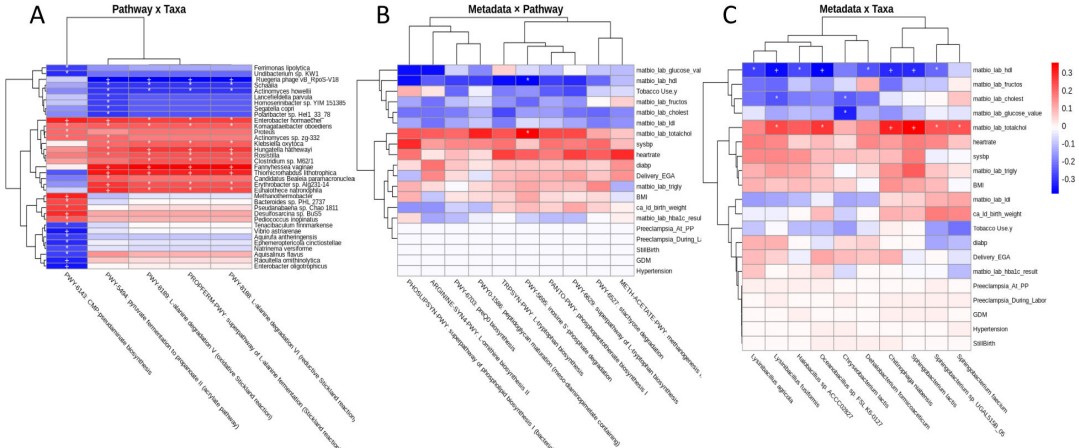

**FIG 5** Spearman correlation heatmaps depicting associations among clinical metadata, differentially abundant taxa, and case-enriched metabolic pathways. (A) Correlations between case-enriched pathways and differentially abundant taxa. (B) Correlations between clinical metadata and differentially abundant taxa. (C) Correlations between clinical metadata and case-enriched pathways. Only taxa significantly different between cases and controls are shown. Colors indicate Spearman correlation coefficients (red, positive; blue, negative), with symbols denoting statistical significance (*P < 0.05; +P < 0.01).

signatures. The corresponding CV-AUC (0.62) and F1-score (0.57) indicated moderate but stable classification performance. The second-best model, SVM on taxa features, attained nearly equivalent generalization (test AUC = 0.80), implying that taxonomic composition alone retained substantial predictive information. Among functional models, XGBoost performed best (test AUC = 0.76, accuracy = 0.73), despite a comparatively lower CV-AUC of 0.54—suggesting effective regularization and strong generalization from functional features alone. The top results for each feature type are reported in Table 2. Slightly higher AUCs observed in the held-out test set relative to the CV-AUC estimates are expected, given the small sample number with high-dimensional features. This reflects refitting on the larger training set and the variance associated with a smaller, single test split.

The aggregated ROC curves revealed overall stable AUC distributions across classifiers and feature sets, illustrated in Fig. 6B. Notably, the combined MLP model achieved a mean test AUC of 0.95 ± 0.06, and KNN reached 0.99 ± 0.02, reflecting exceptionally robust separability under repeated sampling. In contrast, KNN models trained on pathways-only features showed greater performance variability (0.75 ± 0.11), indicating sensitivity to feature sparsity and sample composition. Logistic regression trained on taxa displayed the widest variability (std AUC = 0.11), consistent with its linear nature and lack of feature interaction modeling. Ensemble-based methods (RF, XGB) exhibited the most consistent performance (std ≤ 0.06), highlighting their resilience to small-sample variation through feature bagging and regularization. Together, these findings suggest that integrating microbial taxonomy and function within nonlinear frameworks enhances both predictive stability and interpretability.

To dissect feature-level predictive contributions, we next computed univariate AUCs for all features within each data set, identifying the top 10 most discriminative features per feature set (Fig. 7). Among taxa, *Enterobacter hormaechei* emerged as the single most predictive species (AUC = 0.83), showing marked enrichment in controls ($\log_2$FC = −3.66, FDR = 2.9 × 10$^{-16}$; Fig. 4A). In the functional domain, the acrylate pathway ("PWY-5494: pyruvate fermentation to propanoate II") showed an AUC of 0.72, also enriched in controls ($\log_2$FC = 0.85, FDR = 0.049; Fig. 4B). Interestingly, these two features were positively correlated ($r = 0.38$; Fig. 5) and jointly contributed to the Combined SVM model's decision boundary, suggesting an ecologically coherent microbial network associated with protective metabolic activity.

In the pathway-level domain, amino acid metabolism emerged as a recurring signal, particularly involving L-alanine degradation (PWY-8188, PWY-8189) and the Stickland fermentation pathway (PROPFERM-PWY), both showing high discriminative capacity (AUCs > 0.75). These pathways were also significantly differentially abundant (Fig. 4B), reinforcing their biological relevance. Moreover, the fact that nearly all significantly differentially abundant pathways were also among the top contributors in the XGBoost pathway model (AUCs > 0.69) demonstrates strong coherence between statistical and machine learning–based feature discovery.

Together, these results highlight that disease discrimination is driven not by a single dominant taxon or pathway, but by the joint contribution of functionally linked microbial signatures. The convergence of differential abundance, univariate AUC ranking, and multivariate model weights suggests a biologically grounded set of candidate biomarkers. These taxa–pathway relationships underscore the potential of integrative microbiome-based predictive modeling frameworks to capture disease-relevant microbial ecology beyond univariate associations.

## DISCUSSION

### Community structure in gestational disease

This study provides a high-resolution characterization of the maternal gut microbiome in pregnancies complicated by GH and GDM compared with healthy controls. Using shotgun metagenomics, we observed no significant differences in global beta diversity, but case samples exhibited nominal reductions in both taxonomic and functional alpha

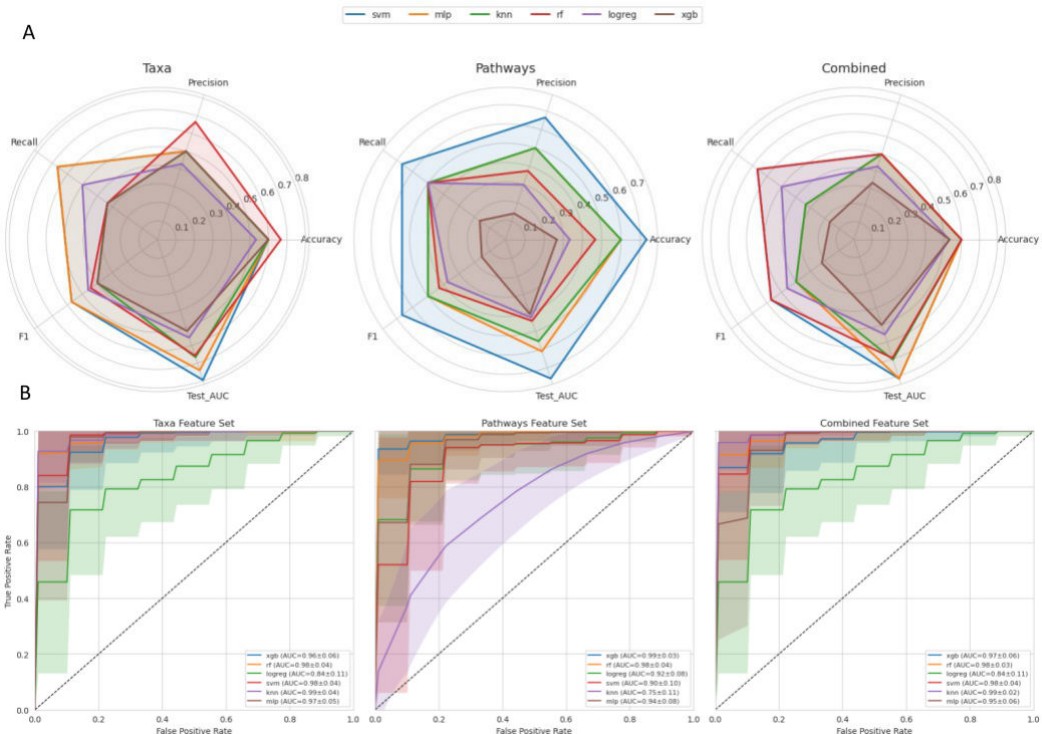

**FIG 6** (A) Radar plots comparing model performance across five evaluation metrics (Accuracy, Precision, Recall, F1 Score, and Test AUC) for each feature set: Taxa, Pathways, and Combined. Each polygon represents a distinct machine learning classifier, allowing direct comparison of predictive performance across both models and feature types. Notably, SVM and MLP achieved consistently high Test AUCs on Combined and Taxa feature sets, while XGBoost excelled on Pathways. (B) Receiver Operating Characteristic (ROC) curves aggregated over 100 stratified test-set iterations, with shaded regions representing ±1 standard deviation. Models demonstrated generally stable AUC estimates, with combined feature sets showing the highest reproducibility across classifiers.

diversity. Although BMI differed between groups, the absence of significant differences in global community metrics such as the Firmicutes-to-Bacteroidetes ratio suggests that BMI alone does not explain the observed disease-associated microbial shifts (Table S3); nevertheless, residual confounding by host factors cannot be fully excluded. Collectively, these findings are consistent with prior reports of microbial depletion in gestational disease and suggest that dysbiosis may be driven less by broad community restructuring and more by selective loss of protective taxa and functional capacity (8, 35, 37).

## Taxonomic alterations suggest disease-associated microbial signatures

At the taxonomic level, we observed depletion of commensals such as *Akkermansia muciniphila*, *Faecalibacterium prausnitzii*, and *Roseburia intestinalis*, alongside enrichment of *Collinsella aerofaciens*, *Veillonella parvula*, and *Raoultella planticola*. The loss of SCFA-producing and anti-inflammatory organisms is consistent with disrupted metabolic signaling and heightened immune activation, both of which are hallmarks of GDM and GH pathophysiology. Interestingly, taxa often considered beneficial, such as *Bifidobacterium longum* and *Lacticaseibacillus rhamnosus*, were elevated in cases. This apparent discrepancy may reflect strain-level heterogeneity or context-dependent functional effects in the altered metabolic and inflammatory environment associated with

**TABLE 2** Top-performing models and their evaluation metrics across feature sets

| Model | Feature set | CV-AUC | Test AUC | Accuracy | F1 score |
|-------|-------------|--------|----------|----------|----------|
| SVM | Combined | 0.62 | 0.81 | 0.60 | 0.57 |
| SVM | Taxa | 0.62 | 0.80 | 0.60 | 0.57 |
| XGB | Pathways | 0.54 | 0.76 | 0.73 | 0.67 |

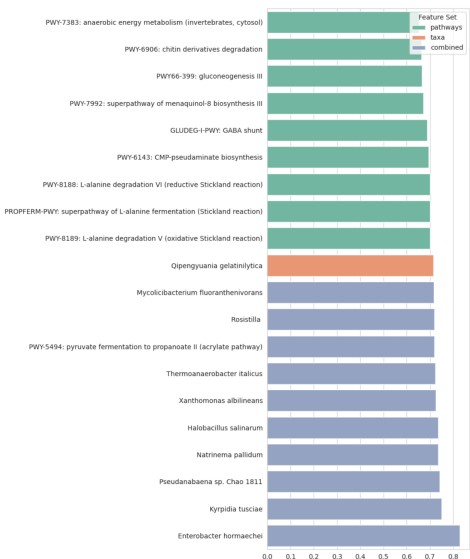

**FIG 7** Top features ranked by univariate AUC across Taxa, Pathways, and Combined feature sets. Bars represent feature-level AUC scores for binary classification of case/control status. Metabolic pathways related to amino acid and short-chain fatty acid metabolism (e.g., L-alanine degradation, pseudaminate biosynthesis) demonstrated strong discriminatory ability, while specific taxa such as *Enterobacter hormaechei* and *Kyrpidia tusciae* also ranked among the most predictive.

gestational disease. As species-level profiling does not resolve strain-specific effects, future metagenomic analyses incorporating strain-level resolution will be necessary to clarify the functional role of *B. longum* and *L. rhamnosus* in this setting.

## Implication of functional metabolic shifts on host physiology

Functional profiling revealed complementary patterns, with enrichment of amino acid fermentation pathways, particularly those involved in Stickland metabolism, and depletion of the acrylate pathway for propionate production. These findings suggest a metabolic environment characterized by reduced SCFA availability and increased amino acid catabolism. SCFAs such as propionate and butyrate are known to modulate insulin sensitivity, endothelial function, and immune tolerance, all of which are disrupted in GH and GDM (7, 8). While much of the insulin resistance is potentially noted to be mediated by human placental lactogen and human placental growth hormone produced by the placenta, the altered abundance of taxa such as *Akkermansia muciniphila*, *Faecalibacterium prausnitzii*, and *Roseburia intestinalis* in the gut potentially contributes to glucose intolerance (44). The correlation between Stickland pathways and pro-inflammatory taxa further supports the idea that altered microbial metabolism may exacerbate systemic inflammation and metabolic stress during pregnancy.

## Predictive modeling reinforces microbial biomarker potential

Beyond descriptive associations, our predictive modeling demonstrates the translational potential of microbiome-derived features. Combined taxonomic and functional abundance inputs yielded the strongest classifiers, with support vector machine and extreme gradient boosted models achieving test AUCs above 0.80. Notably, top features identified through machine learning overlapped with those highlighted by differential abundance testing, including *Enterobacter hormaechei* and the acrylate pathway. While the acrylate pathways are relevant to metabolic stress, *Enterobacter hormaechei* and the other identified taxa in Fig. 3A are known clinical pathogens, further revealing how disease states are both characterized by and vulnerable to opportunistic bacteria. This concordance suggests that the most predictive features are also biologically relevant in

addition to being reproducible, strengthening their candidacy as biomarkers. While our cohort size was modest, the consistent contribution of certain taxa and pathways across both univariate and multivariate frameworks indicates their robustness as potential indicators of gestational disease.

Taken together, these results highlight a model of gestational disease–associated dysbiosis marked by selective loss of beneficial commensals, enrichment of inflammatory and metabolically disruptive taxa, and a functional shift toward amino acid fermentation at the expense of SCFA production. Importantly, our data suggest that microbiome-derived signatures can be harnessed for disease prediction, potentially enabling earlier diagnosis or risk stratification in clinical settings.

## Limitations

Nevertheless, several limitations should be acknowledged. While our study was a planned pilot secondary analysis, it was still limited by a relatively small sample size. The relatively small cohort size limits statistical power to detect subtle microbiome differences, may contribute to false-negative findings, and inflate variance in predictive models, despite the use of repeated train–test split validation. Our cross-sectional design precludes assessment of temporal dynamics, making it unclear whether observed shifts precede disease onset, reflect downstream consequences, or denote trimester-specific alterations as well. The sample size also limited some of the ability to control for some key sociodemographic factors associated with these two adverse outcomes. Pooling data with other microbiome and adverse pregnancy outcome studies could be useful for developing a generalizable profile for predictive modeling. Our cohort was enriched for risk factors for GDM. While several GDM and GH risk factors overlap, validating our findings in an unselected, generalizable pregnancy cohort will be important if this is to be a tool for general screening. Similarly, we will need to validate the predictive model in a separate population. While we collected dietary information and intend to include it in future work, it was not collected consistently in the cohort and thus could not be incorporated into the current analysis. We will have a validated and more detailed dietary index collected in future work to overcome this limitation and potential confounding factor. Additionally, in future work, we plan to assess for microbiome changes during pregnancy and the relationship of abundant species changes to GDM and GH.

## Future directions

Overall, these findings support the hypothesis that gut microbiome alterations contribute to the metabolic and inflammatory disruptions observed in GDM and GH. Given the differences in several key species seen in this study, it is possible that microbiome analysis could potentially be a relatively early biomarker for these adverse outcomes. Additional studies to incorporate microbiome findings into other clinical, genetic, and biochemical factors to develop a robust and clinically useful predictive tool for GH and GDM are planned. In particular, future work will integrate untargeted metabolomics profiling of microbially derived metabolites in maternal stool, blood, and urine samples, enabling direct linkage of microbial functional potential to downstream metabolic activity and host exposure. Functional validation, particularly of SCFA-related pathways, will be critical to disentangle causality. If our findings are validated and replicated in that microbiome changes are independent predictors of adverse outcomes, future research may also explore the therapeutic potential of microbiome-targeted interventions in mitigating the risks associated with these pregnancy-related conditions. Understanding the mechanistic links between microbial composition, metabolic pathways, and host immune responses may pave the way for novel strategies to improve maternal and fetal health outcomes.

## AUTHOR AFFILIATIONS

[1]Department of Computer Science, Indiana University Bloomington, Bloomington, Indiana, USA

<sup>2</sup>Department of Obstetrics and Gynecology, Indiana University School of Medicine, Indianapolis, Indiana, USA
<sup>3</sup>Department of Computer Science, Northeastern University, Boston, Massachusetts, USA

## AUTHOR ORCIDs

Genevieve A. Mortensen http://orcid.org/0009-0003-4370-3646
David M. Haas http://orcid.org/0000-0002-8379-0743

## AUTHOR CONTRIBUTIONS

Genevieve A. Mortensen, Formal analysis, Methodology, Resources, Software, Validation, Visualization, Writing – original draft, Writing – review and editing | Haley Schmidt, Resources, Writing – original draft | Yuzhen Ye, Conceptualization, Data curation, Funding acquisition, Investigation, Project administration, Resources, Supervision | David M. Haas, Conceptualization, Data curation, Funding acquisition, Investigation, Project administration, Resources, Supervision.

## DATA AVAILABILITY

All metagenomes and sample metadata are available at the NCBI Sequence Read Archive under accession BioProject PRJNA1247940. The underlying code for this study is publicly available on GitHub under the repository MGPipe via https://github.com/ginnymortensen/MGPipe.git.

## ADDITIONAL FILES

The following material is available online.

### Supplemental Material

**Supplemental material (Spectrum03155-25-S0001.pdf).** Fig. S1 to S12; Tables S1 and S2.

### Open Peer Review

**PEER REVIEW HISTORY (review-history.pdf).** An accounting of the reviewer comments and feedback.

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
