## [Reviewer comments · Microbiology Spectrum]

Microbiology Spectrum

Metagenomic profiling and predictive modeling of the gut microbiome reveal signatures of gestational disease

Genevieve Mortensen, Haley Schmidt, Predrag Radivojac, Yuzhen Ye, and David Haas

Corresponding Author(s): David Haas, Indiana University School of Medicine

Review Timeline:

Submission Date:	October 21, 2025
Editorial Decision:	December 12, 2025
Revision Received:	February 11, 2026
Accepted:	February 21, 2026

Editor: Ruixin Zhu

Reviewer(s): Disclosure of reviewer identity is with reference to reviewer comments included in decision letter(s). The following individuals involved in review of your submission have agreed to reveal their identity: Lianmin Chen (Reviewer #2); Hongju Chen (Reviewer #3); Jian Zhou (Reviewer #4)

Transaction Report:

DOI: <https://doi.org/10.1128/spectrum.03155-25>

Re: Spectrum03155-25 (Metagenomic profiling and predictive modeling of the gut microbiome reveal signatures of gestational disease)

Dear Dr. David Haas:

Thank you for the privilege of reviewing your work. Below you will find my comments, instructions from the Spectrum editorial office, and the reviewer comments.

Revision Guidelines

Sincerely,
Ruixin Zhu
Editor
Microbiology Spectrum

Reviewer #2 (Comments for the Author):

This study examines how the gut microbiome differs in pregnant women with gestational hypertension (GH) and gestational diabetes mellitus (GDM) compared with healthy controls. Metagenomic sequencing revealed that GH and GDM cases had more variable and distinctly altered microbial communities, including reduced levels of beneficial species such as *Bacteroides fragilis* and *Roseburia intestinalis*. Several *Sphingobacterium* species were differentially abundant, and functional analyses indicated

disruptions in carbohydrate and lipid metabolism. Machine learning models achieved high predictive performance in distinguishing cases from controls, suggesting that gut microbiome profiles may serve as potential biomarkers. Overall, the findings support a significant role for the gut microbiome in pregnancy health and identify it as a promising target for future interventions. However, several major concerns should be addressed:

1. The authors combined GH and GDM into a single "cases" group, which reduces scientific rigor. GH and GDM are distinct conditions; I recommend separating samples into three groups (control, GH, GDM) and re-running the analyses. If the aim is to identify shared microbial signatures, overlap analyses can then be performed on the separate group-specific results.
2. The manuscript is not especially novel in terms of sample size or the reported differences in microbial diversity and abundance, given prior studies on the topic as introduced also in the introduction. I suggest deeper analyses to increase novelty and biological insight. Examples include: microbial co-occurrence / interaction network analysis to test for ecological dysbiosis (for example PMID: 40027484), and targeted or untargeted metabolomics on plasma and fecal samples to strengthen functional interpretation. Adding one or more of these layers would broaden the study's scope and increase its novelty and impact.
3. The prediction approach is not described in the Methods section.
4. Be cautious about model performance claims, without independent replication, results may be optimistic. The authors should validate their models on an independent cohort or apply the model to publicly available datasets for external validation.

Reviewer #3 (Comments for the Author):

Review Comments

This manuscript focuses on a clinically relevant topic—the role of the gut microbiome in gestational hypertension (GH) and gestational diabetes mellitus (GDM). Its use of shotgun metagenomic sequencing combined with machine learning for predictive modeling constitutes a reasonable methodological approach to exploring microbial signatures related to these pregnancy complications. The core findings regarding microbial variability, changes in beneficial taxa, and functional metabolic shifts provide some biological insights into the link between gut dysbiosis and GH/GDM.

Overall, the work bears relevance to maternal-fetal health research and may meet the publication criteria after targeted revisions to improve its rigor and clarity. Specific revision suggestions are as follows:

Major Revision Suggestions

1. Some key data are missing, making it difficult for readers to clearly trace the derivation process and basis of the conclusions. It is recommended that the relevant data information be supplemented and completed.
2. No technical roadmap or analysis flow chart for data processing and data analysis is presented in the paper. Supplementing this visualization will help readers understand the technical framework and analytical logic of the research more intuitively and efficiently.
3. In this study, 73 sample data were divided into a training set and a test set at a ratio of 80%:20%. It should be noted that relying solely on a single stratified experiment to identify differential species would compromise the robustness of the conclusions drawn. In addition, when conducting prediction analysis using six machine learning models, it is advisable to increase the number of repeated predictions and calculate the average AUC value, so as to improve the reliability and persuasiveness of the experimental results.

Minor Revision Suggestions

1. No significant difference was observed in the gut microbiota structure between the Case group and the Control group. It is recommended that the Stress value of Figure 1 (B) (NMDS, non-metric multidimensional scaling analysis) be explicitly presented in the figure or its legend. In addition to reporting the p-value ($p=0.59$), it would be more rigorous to clarify whether the Stress value is less than 0.1, which is critical for validating the reliability of the NMDS ordination results.
2. At the species level, all three-diversity metrics were significantly higher in controls compared to cases (Please provide the specific values of the three-diversity metrics and their corresponding p-values), with p-values $< 2.2 \times 10^{-16}$ in both tests. Similarly, pathway-level richness and diversity were also significantly reduced in the case group (Observed: $\#p = 4.11 \times 10^{-12}$, $\$p = 1.56 \times 10^{-12}$; Shannon: $\#p = 9.18 \times 10^{-6}$, $\$p = 2.86 \times 10^{-5}$; Simpson: $\#p = 6.89 \times 10^{-6}$, $\$p = 4.96 \times 10^{-5}$) (The data presented here are inconsistent with those shown in the corresponding figure; please check and reconcile the discrepancies). These findings indicate a consistent depletion of both taxonomic and functional diversity in gestational disease, consistent with previous findings (Please supplement supporting data or relevant evidence from published studies).
3. Distinct differences in *Akkermansia muciphila*, *Bifidobacterium longum*, *Collinsella aerofaciens*, *Faecalibacterium prausnitzii* and *Roseburia intestinalis* abundance proportions are observed between groups. (please provide corresponding visualization figures and statistical data to verify this claim)
4. Overall, genus-level community composition appeared broadly similar between groups; however, subtle group-specific differences in relative abundance were evident. (please provide corresponding visualization figures and statistical data to support this conclusion).

Reviewer #4 (Comments for the Author):

This study aims to investigate the taxonomic and functional differences of the gut microbiome between pregnant women with gestational hypertension (GH)/gestational diabetes mellitus (GDM) and healthy controls using shotgun metagenomic sequencing combined with machine learning techniques. The key findings include: significantly reduced taxonomic and functional alpha diversity of the gut microbiome in the case group; decreased abundance of beneficial commensals (e.g., *Akkermansia muciniphila*, *Faecalibacterium prausnitzii*) and enrichment of pro-inflammatory or metabolically disruptive bacteria (e.g., *Collinsella aerofaciens*, *Veillonella parvula*); enhanced amino acid fermentation pathways and weakened short-chain fatty acid (SCFA) synthesis pathways at the functional level; and the machine learning model (e.g., linear SVM) integrating taxonomic and functional features achieved an AUC of 0.81 for disease prediction, demonstrating the potential of gut microbes as biomarkers for pregnancy-related diseases.

The core contribution of this study lies in breaking through the resolution limitation of traditional 16S rRNA sequencing, revealing the gut microbiome taxonomic-functional synergistic characteristics associated with GH/GDM through high-resolution metagenomic technology, and developing a reproducible bioinformatics pipeline MGPipe to provide technical support for subsequent studies. The results provide a new perspective for early screening, risk stratification, and microbiome-targeted interventions of pregnancy-related diseases. However, there are still deficiencies in sample representativeness, causal relationship verification, and confounding factor control, which need to be further improved to enhance the robustness and translational value of the study.

Questions and Rebuttal Suggestions for Authors

Regarding the small sample size, have you performed a statistical power analysis to clarify the ability of the current sample size to detect microbiome differences? If not, can you supplement the discussion on the potential impact of insufficient sample size on the research conclusions?

The cross-sectional design cannot establish a causal relationship. Do you plan to conduct a longitudinal follow-up study to track the microbiome changes throughout pregnancy? Or can you provide indirect evidence for the causal association by supplementing the correlation analysis between microbial characteristics and disease severity?

Diet is a key factor regulating the gut microbiome. Why was dietary data not included in the analysis of this study? Will you integrate dietary information in the future to exclude its confounding effect on the microbe-disease association?

High-risk populations for GDM were prioritized during recruitment. How does this design affect the balance of baseline characteristics of the cohort? How do you plan to validate the study results in the general pregnant population to improve its clinical universality?

The enrichment of traditionally beneficial bacteria such as *Bifidobacterium longum* in the case group is contradictory to existing studies. Have you explored the impact of strain-level differences (e.g., through metagenome-assembled genomes analysis) or changes in the host metabolic environment on the functional roles of these bacteria?

The machine learning model has good predictive performance but lacks external validation. Do you have access to an independent cohort for model validation? Can you further explain the biological mechanism by which core predictive features (e.g., *Enterobacter hormaechei*, PWY-5494 pathway) participate in the pathogenesis of GH/GDM?

Is there a correlation between differentially enriched functional pathways (e.g., Stickland metabolism, acrylate pathway) and clinical indicators of patients (e.g., blood glucose, blood pressure)? Supplementing such analysis can enhance the clinical relevance of functional findings. Do you consider supplementing relevant data?

Can you supplement the quality control details of metagenomic data, such as the average sequencing depth of each sample, the efficiency of host DNA removal, and the filtering threshold of low-abundance taxa? How do these parameters affect the results of taxonomic and functional profiling?

After correcting for batch effects, were other potential confounding variables such as BMI and parity adjusted? If not, how might these variables affect the observed microbial differential characteristics?

The authors mentioned that the MGPipe pipeline is reproducible. Can you provide a detailed operation tutorial or supplementary documentation to facilitate the use of other researchers? Has the compatibility of the pipeline in different sequencing platforms or cohorts of different sizes been tested?

Review Comments

This manuscript focuses on a clinically relevant topic—the role of the gut microbiome in gestational hypertension (GH) and gestational diabetes mellitus (GDM). Its use of shotgun metagenomic sequencing combined with machine learning for predictive modeling constitutes a reasonable methodological approach to exploring microbial signatures related to these pregnancy complications. The core findings regarding microbial variability, changes in beneficial taxa, and functional metabolic shifts provide some biological insights into the link between gut dysbiosis and GH/GDM.

Overall, the work bears relevance to maternal-fetal health research and may meet the publication criteria after targeted revisions to improve its rigor and clarity. Specific revision suggestions are as follows:

Major Revision Suggestions

1. Some key data are missing, making it difficult for readers to clearly trace the derivation process and basis of the conclusions. It is recommended that the relevant data information be supplemented and completed.
2. No technical roadmap or analysis flow chart for data processing and data analysis is presented in the paper. Supplementing this visualization will help readers understand the technical framework and analytical logic of the research more intuitively and efficiently.
3. In this study, 73 sample data were divided into a training set and a test set at a ratio of 80%:20%. It should be noted that relying solely on a single stratified experiment to identify differential species would compromise the robustness of the conclusions drawn. In addition, when conducting prediction analysis using six machine learning models, it is advisable to increase the number of repeated predictions and calculate the average AUC value, so as to improve the reliability and persuasiveness of the experimental results.

Minor Revision Suggestions

1. No significant difference was observed in the gut microbiota structure between the Case group and the Control group. It is recommended that the **Stress value** of Figure 1 (B) (NMDS, non-metric multidimensional scaling analysis) be explicitly presented in the figure or its legend. In addition to reporting the p-value ($p=0.59$), it would be more rigorous to clarify whether the Stress value is less than 0.1, which is critical for validating the reliability of the NMDS ordination results.
2. At the species level, all three-diversity metrics were significantly higher in controls compared to cases (**Please provide the specific values of the three-diversity metrics and**

their corresponding p-values), with p-values $< 2.2 \times 10^{-16}$ in both tests. Similarly, pathway-level richness and diversity were also significantly reduced in the case group (Observed: #p = 4.11×10^{-12} , \$p = 1.56×10^{-12} ; Shannon: #p = 9.18×10^{-6} , \$p = 2.86×10^{-5} ; Simpson: #p = 6.89×10^{-6} , \$p = 4.96×10^{-5}) **(The data presented here are inconsistent with those shown in the corresponding figure; please check and reconcile the discrepancies)**. These findings indicate a consistent depletion of both taxonomic and functional diversity in gestational disease, consistent with previous findings **(Please supplement supporting data or relevant evidence from published studies)**.

3. Distinct differences in Akkermansia municipihila, Bifidobacterium longum, Collinsella aerofaciens, Faecalibacterium prausnitzii and Roseburia intestinalis abundance proportions are observed between groups. **(please provide corresponding visualization figures and statistical data to verify this claim)**

4. Overall, genus-level community composition appeared broadly similar between groups; however, subtle group-specific differences in relative abundance were evident. (please provide corresponding visualization figures and statistical data to support this conclusion).

Response to Reviewers

Rebuttal comments are italicized

Reviewer #2 (Comments for the Author):

This study examines how the gut microbiome differs in pregnant women with gestational hypertension (GH) and gestational diabetes mellitus (GDM) compared with healthy controls. Metagenomic sequencing revealed that GH and GDM cases had more variable and distinctly altered microbial communities, including reduced levels of beneficial species such as *Bacteroides fragilis* and *Roseburia intestinalis*. Several *Sphingobacterium* species were differentially abundant, and functional analyses indicated disruptions in carbohydrate and lipid metabolism. Machine learning models achieved high predictive performance in distinguishing cases from controls, suggesting that gut microbiome profiles may serve as potential biomarkers. Overall, the findings support a significant role for the gut microbiome in pregnancy health and identify it as a promising target for future interventions. However, several major concerns should be addressed:

1. The authors combined GH and GDM into a single "cases" group, which reduces scientific rigor. GH and GDM are distinct conditions; I recommend separating samples into three groups (control, GH, GDM) and re-running the analyses. If the aim is to identify shared microbial signatures, overlap analyses can then be performed on the separate group-specific results.

We agree with the reviewer's point that GH and GDM are distinct conditions. Because a subset of participants met criteria for both diagnoses and subgroup sample sizes were limited, we conducted primary analyses using a combined case group to maximize statistical power and identify shared microbial alterations. We performed stratified sensitivity analyses comparing healthy controls to GH and GDM separately by calculating alpha and beta diversities, differential abundance, and overlap assessment of significant microbial signatures. These results are now presented in the Supplementary Figures 7-12 and support the overall conclusions of the study. This has been explained in the Methods section under subsection "Study design and data description": "GH and GDM often co-occur, and among the 18 women with GDM, 11 also had GH. Given the partial diagnostic overlap between GH and GDM and the limited subgroup sample sizes, we performed the primary analyses using a combined case group. In addition, stratified analyses comparing controls to GH and GDM separately were conducted as sensitivity analyses and are provided in Supplementary Figures 7 - 12. These analyses revealed shared microbial markers between the GH and GDM groups (see Results), further supporting our rationale for the combined analysis." The specific results of the overlap analysis are mentioned in the Results section under subsection "Diseased gut microbiome diversity and composition differs from the healthy gut during pregnancy": "Stratified analyses revealed largely overlapping microbial signatures in GH and GDM relative to controls. Of the 8 statistically significant differentially abundant microbial species in the GDM cohort, 5 of them are also differentially abundant in the GH cohort with a threshold of $p < 0.05$ (Supplementary Figures 7 - 12). This result supports the rationale for the combined case analysis."

2. The manuscript is not especially novel in terms of sample size or the reported differences in microbial diversity and abundance, given prior studies on the topic as introduced also in the introduction. I suggest deeper analyses to increase novelty and biological insight. Examples include: microbial co-occurrence / interaction network analysis to test for ecological dysbiosis (for example PMID: 40027484), and targeted or untargeted metabolomics on plasma and fecal samples to strengthen functional interpretation. Adding one or more of these layers would broaden the study’s scope and increase its novelty and impact.

*We have added differentially significant co-abundant overlap analyses to Supplementary Figure 12 per PMID 40027484 as the reviewer suggests: "Using a significance threshold of $p < 0.05$, we identify the taxa which are significantly differentially abundant in GDM and GH cohorts independently relative to healthy controls. Of these, 5 microbial species were identified as differentially abundant and statistically significant taxa in both disease cohorts. These species were *Dehalobacterium formicoaceticum*, *Lysinibacillus agricola*, *Paenibacillus* sp. FSL E2-0178, *Sphingobacterium faecium* and *Chitinophaga niabensis*. These taxa reinforce the results seen in our combined cohort in the main text, however, *Dehalobacterium formicoaceticum*, *Paenibacillus* sp. FSL E2-0178 and *Chitinophaga niabensis* are not highly abundant and are likely identified due to the small sample size when separating groups based on condition, with preceding analyses showing low statistical power in separated cohorts." We have also added this reference to the Introduction section: "Moreover, there is a paucity of well-controlled studies that directly compare both taxonomic and functional features or their overlap across GH, GDM, and healthy pregnancies using high-resolution metagenomic data" While the inclusion of metabolomic analyses is beyond the scope of the current manuscript, these analyses are already planned as part of an ongoing, funded study and is now added to the Discussion section under the subsection Future Directions: "Additional studies to incorporate microbiome findings into other clinical, genetic, and biochemical factors, to develop a robust and clinically useful predictive tool for GH and GDM are planned. In particular, future work will integrate untargeted metabolomics profiling of microbially derived metabolites in maternal stool, blood, and urine samples, enabling direct linkage of microbial functional potential to downstream metabolic activity and host exposure. "*

3. The prediction approach is not described in the Methods section.

*The prediction approach has been separated from the Results section and moved to the Methods section, under subsection Predictive modeling framework: "To assess the predictive value of microbial features, we trained and evaluated a panel of six classifiers—XGBoost (XGB), Random Forest (RF), Logistic Regression (LR), Support Vector Machine (SVM), K-Nearest Neighbors (KNN), and Multi-Layer Perceptron (MLP)—on three different input feature sets: taxonomic profiles (X_{taxa}), functional pathway profiles ($X_{pathways}$), and their combination ($X_{combined}$)—to create a total of 18 distinct predictive models. Each model was implemented in **scikit-learn** using standardized supervised learning workflows. Each profile represented feature-level attributes including relative abundance, false discovery rate, and \log_2 fold change, concatenated to capture both quantitative and statistical significance information. Models were trained to predict the binary disease label vector y , corresponding to each subject’s disease-state status. Given the small sample size ($n = 73$; 57% cases and 43% controls), we employed a stratified 80/20 train/test split (random seed 42) to preserve class proportions and ensure balanced evaluation. Within the training subset, 5-fold cross-*

validation with grid search was used to tune model-specific hyperparameters, optimizing the Area Under the Receiver Operating Characteristic Curve (AUC) as the performance criterion. The final optimized models were then evaluated on the held-out test set, reporting accuracy, precision, recall, F1-score, and test AUC. To further assess generalization and model robustness, we conducted Monte Carlo cross-validation by re-sampling 100 stratified train/test splits (seeds 0–99) and applying each previously trained model to new 20% test subsets. For each iteration, ROC curves were computed and interpolated over a common false positive rate (FPR) grid to derive mean true positive rates (TPRs) and standard deviation bands, producing smoothed ROC curves with confidence intervals. This resampling approach quantifies how consistently models discriminate under varying data partitions as a critical consideration in small-cohort microbiome studies prone to sampling noise.”

4. Be cautious about model performance claims, without independent replication, results may be optimistic. The authors should validate their models on an independent cohort or apply the model to publicly available datasets for external validation.

We added a part to the limitations that the predictive model will also need to be replicated in an independent cohort: ”While several GDM and GH risk factors overlap, validating our findings in an unselected, generalizable pregnancy cohort will be important if this is to be a tool for general screening. Similarly, we will need to validate the predictive model in a separate population. ”

Reviewer #3 (Comments for the Author):

Review Comments

This manuscript focuses on a clinically relevant topic—the role of the gut microbiome in gestational hypertension (GH) and gestational diabetes mellitus (GDM). Its use of shotgun metagenomic sequencing combined with machine learning for predictive modeling constitutes a reasonable methodological approach to exploring microbial signatures related to these pregnancy complications. The core findings regarding microbial variability, changes in beneficial taxa, and functional metabolic shifts provide some biological insights into the link between gut dysbiosis and GH/GDM.

Overall, the work bears relevance to maternal-fetal health research and may meet the publication criteria after targeted revisions to improve its rigor and clarity. Specific revision suggestions are as follows:

Major Revision Suggestions

1. Some key data are missing, making it difficult for readers to clearly trace the derivation process and basis of the conclusions. It is recommended that the relevant data information be supplemented and completed.

All data, including metadata and sequencing data, is completed and its accessibility is provided in the Data Availability statement, through <https://www.ncbi.nlm.nih.gov/bioproject/PRJNA1247940>: ”All metagenomes and sample metadata are available at the NCBI Sequence Read Archive under accession BioProject PRJNA1247940 via <https://www.ncbi.nlm.nih.gov/bioproject/PRJNA1247940>.”

//www.ncbi.nlm.nih.gov/bioproject/PRJNA1247940. The underlying code for this study is publicly available on GitHub under the repository MGPipe via <https://github.com/ginnymortensen/MGPipe.git>.

2. No technical roadmap or analysis flow chart for data processing and data analysis is presented in the paper. Supplementing this visualization will help readers understand the technical framework and analytical logic of the research more intuitively and efficiently.

We have added this roadmap describing the workflow of the data, from processing to analysis, to Supplementary Figure 6: "Overview of the metagenomic data processing, statistical analysis, and predictive modeling workflow. Stool samples were subjected to standardized shotgun metagenomic sequencing and processed using the MGPipe bioinformatics pipeline, followed by taxonomic and functional profiling, batch correction, differential abundance testing, and supervised machine learning analyses."

3. In this study, 73 sample data were divided into a training set and a test set at a ratio of 80%:20%. It should be noted that relying solely on a single stratified experiment to identify differential species would compromise the robustness of the conclusions drawn. In addition, when conducting prediction analysis using six machine learning models, it is advisable to increase the number of repeated predictions and calculate the average AUC value, so as to improve the reliability and persuasiveness of the experimental results.

This approach was employed and is now described in the Methods section: "Given the small sample size ($n = 73$; 57% cases and 43% controls), we employed a stratified 80/20 train/test split (random seed 42) to preserve class proportions and ensure balanced evaluation. Within the training subset, 5-fold cross-validation with grid search was used to tune model-specific hyperparameters, optimizing the Area Under the Receiver Operating Characteristic Curve (AUC) as the performance criterion. The final optimized models were then evaluated on the held-out test set, reporting accuracy, precision, recall, F1-score, and test AUC. To further assess generalization and model robustness, we conducted Monte Carlo cross-validation by re-sampling 100 stratified train/test splits (seeds 0–99) and applying each previously trained model to new 20% test subsets. For each iteration, ROC curves were computed and interpolated over a common false positive rate (FPR) grid to derive mean true positive rates (TPRs) and standard deviation bands, producing smoothed ROC curves with confidence intervals. This resampling approach quantifies how consistently models discriminate under varying data partitions as a critical consideration in small-cohort microbiome studies prone to sampling noise." It is also presented in Figure 6B of the Results section: "Receiver Operating Characteristic (ROC) curves aggregated over 100 stratified test-set iterations, with shaded regions representing ± 1 standard deviation. Models demonstrated generally stable AUC estimates, with Combined feature sets showing the highest reproducibility across classifiers."

Minor Revision Suggestions

1. No significant difference was observed in the gut microbiota structure between the Case group and the Control group. It is recommended that the Stress value of Figure 1 (B) (NMDS, non-metric multidimensional scaling analysis) be explicitly presented in the figure

or its legend. In addition to reporting the p-value ($p=0.59$), it would be more rigorous to clarify whether the Stress value is less than 0.1, which is critical for validating the reliability of the NMDS ordination results.

This has been added to the legend of Figure 1B: "NMDS ordination using the same data, with 95% confidence ellipses. MANOVA applied to the first two NMDS axes yielded no significant group effect ($p = 0.59$, stress = 0.13)." It is also clarified in the preceding paragraph: "Similarly, MANOVA applied to the first two NMDS axes yielded a non-significant result ($p = 0.59$). The NMDS ordination showed an acceptable representation of the data in two dimensions (stress = 0.13), suggesting no gross shifts in beta diversity due to gestational disease status."

2. At the species level, all three-diversity metrics were significantly higher in controls compared to cases (Please provide the specific values of the three-diversity metrics and their corresponding p-values), with p-values $\leq 2.2 \times 10^{-16}$ in both tests. Similarly, pathway-level richness and diversity were also significantly reduced in the case group (Observed: $\#p = 4.11 \times 10^{12}$, $\$p = 1.56 \times 10^{12}$; Shannon: $\#p = 9.18 \times 10^6$, $\$p = 2.86 \times 10^5$; Simpson: $\#p = 6.89 \times 10^6$, $\$p = 4.96 \times 10^5$) (The data presented here are inconsistent with those shown in the corresponding figure; please check and reconcile the discrepancies). These findings indicate a consistent depletion of both taxonomic and functional diversity in gestational disease, consistent with previous findings (Please supplement supporting data or relevant evidence from published studies).

These discrepancies have been reconciled and findings have been supplemented with relevant evidence from published studies in the paragraph preceding Figure 2: "Violin plots were generated to visualize group-level distributions of each metric. Linear models (`lm()`) and non-parametric Mann-Whitney U tests (`wilcox.test()`) were used to test for differences in alpha diversity. At the species level, all three diversity metrics were nominally higher in controls compared to cases, but the effect did not remain after adjustment (Observed: $\#p = 4.65 \times 10^{-2}$, $\$p = 9.75 \times 10^{-2}$; Shannon: $\#p = 1.16 \times 10^{-1}$, $\$p = 1.28 \times 10^{-1}$; Simpson: $\#p = 2.46 \times 10^{-1}$, $\$p = 2.62 \times 10^{-1}$). Similarly, pathway-level richness and diversity showed no significant differences following adjustment (Observed: $\#p = 4.67 \times 10^{-1}$, $\$p = 5.58 \times 10^{-1}$; Shannon: $\#p = 5.09 \times 10^{-1}$, $\$p = 2.85 \times 10^{-1}$; Simpson: $\#p = 6.09 \times 10^{-1}$, $\$p = 4.88 \times 10^{-1}$). These results suggest that gestational disease status is not associated with large-scale shifts in within-sample microbial diversity, but may instead reflect more subtle compositional or functional changes, consistent with prior reports of heterogeneous alpha diversity findings alongside compositional alterations in gestational diabetes (cited Teixeira et al. 2023, 'The Association between Gestational Diabetes and the Microbiome: A Systematic Review and Meta-Analysis')

3. Distinct differences in *Akkermansia muciphila*, *Bifidobacterium longum*, *Collinsella aerofaciens*, *Faecalibacterium prausnitzii* and *Roseburia intestinalis* abundance proportions are observed between groups. (please provide corresponding visualization figures and statistical data to verify this claim).

These findings are visualized in Figure 3A and explained in its preceding paragraph: "To explore broader taxonomic shifts at the genus level, we plotted the mean relative abundances of the six most dominant genera across case and control groups. Overall, genus-level com-

munity composition appeared broadly similar between groups; however, subtle group-specific differences in relative abundance were evident. For instance, the genus *Alistipes* was modestly enriched in the control group. This finding is consistent with prior studies reporting an inverse relationship between *Alistipes* abundance and gut inflammation, as well as its reduced prevalence in preeclamptic individuals. Conversely, the genus *Segatella* was relatively enriched in the case group, a trend that aligns with prior associations of this taxon with systemic inflammation, impaired glucose tolerance, and metabolic dysfunction. Direct Case/Control comparisons of selected taxa abundances and Firmicutes/Bacteroidetes ratios are presented in Supplementary Figures 2 and 3, respectively.”

4. Overall, genus-level community composition appeared broadly similar between groups; however, subtle group-specific differences in relative abundance were evident. (please provide corresponding visualization figures and statistical data to support this conclusion).

*These findings have been clarified in the paragraph preceding Figure 2: "Violin plots were generated to visualize group-level distributions of each metric (Figure 2A,B). Linear models (lm()) and non-parametric Mann-Whitney U tests (wilcox.test()) were used to test for differences in alpha diversity. At the species level, all three diversity metrics were nominally higher in controls compared to cases, but the effect did not remain after adjustment (Observed: #p = 4.65 × 10⁻², \$p = 9.75 × 10⁻²; Shannon: #p = 1.16 × 10⁻¹, \$p = 1.28 × 10⁻¹; Simpson: #p = 2.46 × 10⁻¹, \$p = 2.62 × 10⁻¹). Similarly, pathway-level richness and diversity showed no significant differences following adjustment (Observed: #p = 4.67 × 10⁻¹, \$p = 5.58 × 10⁻¹; Shannon: #p = 5.09 × 10⁻¹, \$p = 2.85 × 10⁻¹; Simpson: #p = 6.09 × 10⁻¹, \$p = 4.88 × 10⁻¹). These results suggest that gestational disease status is not associated with large-scale shifts in within-sample microbial diversity, but may instead reflect more subtle compositional or functional changes, consistent with prior reports of heterogeneous alpha diversity findings alongside compositional alterations in gestational diabetes." It has also been visualized in Figure 3B and explained in the paragraph preceding Figure 3: "To explore broader taxonomic shifts at the genus level, we plotted the mean relative abundances of the six most dominant genera across case and control groups (Figure 3B). Overall, genus-level community composition appeared broadly similar between groups; however, subtle group-specific differences in relative abundance were evident. For instance, the genus *Alistipes* was modestly enriched in the control group. This finding is consistent with prior studies reporting an inverse relationship between *Alistipes* abundance and gut inflammation, as well as its reduced prevalence in preeclamptic individuals. Conversely, the genus *Segatella* was relatively enriched in the case group, a trend that aligns with prior associations of this taxon with systemic inflammation, impaired glucose tolerance, and metabolic dysfunction. Direct Case/Control comparisons of selected taxa abundances and Firmicutes/Bacteroidetes ratios are presented in Supplementary Figures 2 and 3, respectively."*

Reviewer #4 (Comments for the Author):

This study aims to investigate the taxonomic and functional differences of the gut microbiome between pregnant women with gestational hypertension (GH)/gestational diabetes mellitus (GDM) and healthy controls using shotgun metagenomic sequencing combined with

machine learning techniques. The key findings include: significantly reduced taxonomic and functional alpha diversity of the gut microbiome in the case group; decreased abundance of beneficial commensals (e.g., *Akkermansia muciniphila*, *Faecalibacterium prausnitzii*) and enrichment of pro-inflammatory or metabolically disruptive bacteria (e.g., *Collinsella aerofaciens*, *Veillonella parvula*); enhanced amino acid fermentation pathways and weakened short-chain fatty acid (SCFA) synthesis pathways at the functional level; and the machine learning model (e.g., linear SVM) integrating taxonomic and functional features achieved an AUC of 0.81 for disease prediction, demonstrating the potential of gut microbes as biomarkers for pregnancy-related diseases.

The core contribution of this study lies in breaking through the resolution limitation of traditional 16S rRNA sequencing, revealing the gut microbiome taxonomic-functional synergistic characteristics associated with GH/GDM through high-resolution metagenomic technology, and developing a reproducible bioinformatics pipeline MGPipe to provide technical support for subsequent studies. The results provide a new perspective for early screening, risk stratification, and microbiome-targeted interventions of pregnancy-related diseases. However, there are still deficiencies in sample representativeness, causal relationship verification, and confounding factor control, which need to be further improved to enhance the robustness and translational value of the study.

Questions and Rebuttal Suggestions for Authors

1. Regarding the small sample size, have you performed a statistical power analysis to clarify the ability of the current sample size to detect microbiome differences? If not, can you supplement the discussion on the potential impact of insufficient sample size on the research conclusions?

We have added the impact of insufficient sample size on research conclusions to the limitations section: "While our study was a planned pilot secondary analysis, it was still limited by a relatively small sample size. The relatively small cohort size limits statistical power to detect subtle microbiome differences, may contribute to false-negative findings, and inflate variance in predictive models, despite the use of repeated train-test split validation."

2. The cross-sectional design cannot establish a causal relationship. Do you plan to conduct a longitudinal follow-up study to track the microbiome changes throughout pregnancy? Or can you provide indirect evidence for the causal association by supplementing the correlation analysis between microbial characteristics and disease severity?

In our next obstetric cohort, we plan to collect stool samples at multiple time points. A few of the HMC participants had multiple stool samples but not enough to power an analysis as the reviewer proposes. In the limitations section, we have written: "Our cross-sectional design precludes assessment of temporal dynamics, making it unclear whether observed shifts precede disease onset, reflect downstream consequences, or denote trimester specific alterations as well ... Additionally, in future work we plan to assess for microbiome changes during pregnancy and the relationship of abundant species changes to GDM and GH."

3. Diet is a key factor regulating the gut microbiome. Why was dietary data not included in the analysis of this study? Will you integrate dietary information in the future to exclude

its confounding effect on the microbe-disease association?

Unfortunately, dietary information was not consistently collected in the entire cohort, limiting its ability to be used. We have also added that in the limitations section: "While we collected dietary information and intend to include it in future work, it was not collected consistently in the cohort and thus not able to be incorporated into the current analysis. We will have a validated and more detailed dietary index collected in future work to overcome this limitation and potential confounding factor."

4. High-risk populations for GDM were prioritized during recruitment. How does this design affect the balance of baseline characteristics of the cohort? How do you plan to validate the study results in the general pregnant population to improve its clinical universality?

The details of this are in the methods paper, which has been cited: "Haas et al. 2024, 'Early pregnancy associations with Gestational Diabetes: methods and cohort results of the Hoosier Moms Cohort'. The validation plans are in the discussion now: "Our cohort was enriched for risk factors for GDM. While several GDM and GH risk factors overlap, validating our findings in an unselected, generalizable pregnancy cohort will be important if this is to be a tool for general screening. Similarly, we will need to validate the predictive model in a separate population. While we collected dietary information and intend to include it in future work, it was not collected consistently in the cohort and thus not able to be incorporated into the current analysis. We will have a validated and more detailed dietary index collected in future work to overcome this limitation and potential confounding factor. Additionally, in future work we plan to assess for microbiome changes during pregnancy and the relationship of abundant species changes to GDM and GH."

5. The enrichment of traditionally beneficial bacteria such as *Bifidobacterium longum* in the case group is contradictory to existing studies. Have you explored the impact of strain-level differences (e.g., through metagenome-assembled genomes analysis) or changes in the host metabolic environment on the functional roles of these bacteria?

*We have added this consideration to the Discussion section under the subsection "Taxonomic alterations suggest disease-associated microbial signatures": "At the taxonomic level, we observed depletion of commensals such as *Akkermansia muciniphila*, *Faecalibacterium prausnitzii*, and *Roseburia intestinalis*, alongside enrichment of *Collinsella aerofaciens*, *Veillonella parvula*, and *Raoultella planticola*. The loss of SCFA-producing and anti-inflammatory organisms is consistent with disrupted metabolic signaling and heightened immune activation, both of which are hallmarks of GDM and GH pathophysiology. Interestingly, taxa often considered beneficial, such as *Bifidobacterium longum* and *Lactobacillus rhamnosus*, were elevated in cases. This apparent discrepancy may reflect strain-level heterogeneity or context-dependent functional effects in the altered metabolic and inflammatory environment associated with gestational disease. As species-level profiling does not resolve strain-specific effects, future metagenomic analyses incorporating strain-level resolution will be necessary to clarify the functional role of *B. longum* and *L. rhamnosus* in this setting."*

6. The machine learning model has good predictive performance but lacks external validation. Do you have access to an independent cohort for model validation? Can you further explain the biological mechanism by which core predictive features (e.g., *Enterobacter*

hormaechei, PWY-5494 pathway) participate in the pathogenesis of GH/GDM?

We have added this plan in the limitations section: "Nevertheless, several limitations should be acknowledged. While our study was a planned pilot secondary analysis, it was still limited by a relatively small sample size. The relatively small cohort size limits statistical power to detect subtle microbiome differences, may contribute to false-negative findings, and inflate variance in predictive models, despite the use of repeated train-test split validation. Our cross-sectional design precludes assessment of temporal dynamics, making it unclear whether observed shifts precede disease onset, reflect downstream consequences, or denote trimester specific alterations as well. The sample size also limited some of the ability to control for some key sociodemographic factors associated with these two adverse outcomes. Pooling data with other microbiome and adverse pregnancy outcome studies could be useful for developing a generalizable profile for predictive modeling. Our cohort was enriched for risk factors for GDM. While several GDM and GH risk factors overlap, validating our findings in an unselected, generalizable pregnancy cohort will be important if this is to be a tool for general screening. Similarly, we will need to validate the predictive model in a separate population. While we collected dietary information and intend to include it in future work, it was not collected consistently in the cohort and thus not able to be incorporated into the current analysis. We will have a validated and more detailed dietary index collected in future work to overcome this limitation and potential confounding factor. Additionally, in future work we plan to assess for microbiome changes during pregnancy and the relationship of abundant species changes to GDM and GH."

7. Is there a correlation between differentially enriched functional pathways (e.g., Stickland metabolism, acrylate pathway) and clinical indicators of patients (e.g., blood glucose, blood pressure)? Supplementing such analysis can enhance the clinical relevance of functional findings. Do you consider supplementing relevant data?

We have added this analysis in Figure 5: "Spearman correlation heatmaps depicting associations among clinical metadata, differentially abundant taxa, and case-enriched metabolic pathways. (A) Correlations between case-enriched pathways and differentially abundant taxa. (B) Correlations between clinical metadata and differentially abundant taxa. (C) Correlations between clinical metadata and case-enriched pathways. Only taxa significantly different between cases and controls are shown. Colors indicate Spearman correlation coefficients (red, positive; blue, negative), with symbols denoting statistical significance ($p < 0.05$; + $p < 0.01$)."* We also discuss it briefly in the paragraph preceding Figure 5: "These findings indicate that case samples exhibit coordinated restructuring of the gut microbiome at both taxonomic and functional levels. To evaluate clinical relevance, we correlated differentially abundant taxa and pathways with host clinical measures in Figure 5 B and C. Both taxa and pathways showed the strongest associations with continuous cardiometabolic biomarkers, particularly lipid-related measures. Several taxa, including *Lysinibacillus* and *Sphingobacterium* species, were positively correlated with total cholesterol, triglycerides, BMI, and systolic blood pressure, forming a shared metabolic signature. Similarly, case-enriched pathways involved in amino acid biosynthesis, phospholipid metabolism, and central carbon metabolism were positively associated with adverse lipid profiles, BMI, and heart rate, while glucose and HDL tended to be inversely correlated. In contrast, categorical pregnancy outcomes and clinical diagnoses showed minimal associations. Together, these results suggest that microbiome

alterations in cases are more closely linked to underlying metabolic physiology than to binary disease labels, with additional differential abundance visualizations provided in Supplementary Figures 4 and 5.” We have also added the individual clinical variable with functional pathway correlation heatmap to Supplementary Figure 4. ”To evaluate the clinical relevance of differentially enriched microbial features, Spearman correlation analyses were performed between host clinical indicators and (A) differentially abundant bacterial taxa and (B) differentially enriched functional pathways. Correlation coefficients are shown for each feature–metadata pair. Associations marked with an asterisk (*) indicate nominal significance ($p < 0.05$), while those marked with a plus sign (+) indicate stronger significance ($p < 0.01$). These analyses provide exploratory evidence linking microbial taxonomic and functional variation with host metabolic and clinical parameters. ”

8. Can you supplement the quality control details of metagenomic data, such as the average sequencing depth of each sample, the efficiency of host DNA removal, and the filtering threshold of low-abundance taxa? How do these parameters affect the results of taxonomic and functional profiling?

We have added these details in Supplementary Table 2 : ”Metagenomic sequencing generated a mean of 29.99 million raw reads per sample, with 29.49 million reads retained after quality control (mean retention: 98.13%). Sequencing quality was high across samples (mean $Q30 = 0.92$), and no samples were excluded based on sequencing depth (minimum threshold: 1,000 reads). No residual host genome sequences were detected because the sequencing facility provided data that had already been depleted of host DNA. To reduce noise from sparsely observed features, taxa and pathways were filtered using a prevalence threshold of $\geq 10\%$ of samples. High sequencing depth and prevalence-based filtering improve the stability of taxonomic and functional profiling by minimizing the influence of low-quality reads and rare features.” We also mention this is in the Methods section under subsection ”Sample processing and sequencing”: ”Quality control details of the metagenomic sequencing data are presented in Supplementary Table 2.”

9. After correcting for batch effects, were other potential confounding variables such as BMI and parity adjusted? If not, how might these variables affect the observed microbial differential characteristics?

We have added this to the Discussion section: ”Although BMI differed between groups, the absence of significant differences in global community metrics such as the Firmicutes-to-Bacteroidetes ratio suggests that BMI alone does not explain the observed disease-associated microbial shifts (Supplementary Table 3); nevertheless, residual confounding by host factors cannot be fully excluded”. After batch correction, additional host-level covariates such as BMI and parity were not adjusted for because batch effects were limited to technical variation in taxonomic and functional sequencing profiles. Although BMI differed significantly between case and control groups (Table 1), no significant differences in the Firmicutes-to-Bacteroidetes ratio were observed (Supplementary Figure 3), suggesting that BMI-related global compositional effects are unlikely to account for the observed microbial differences. We acknowledge that residual confounding by host factors can’t be fully excluded, particularly given the limited sample size. In Supplementary Figure 3: ”This figure displays the ratio of Firmicutes to Bacteroidetes as a standardized way of compared microbial consis-

cies between two groups. We take the raw abundances of relevant taxon within each phyla, Winsorize outliers, then calculate mean F/B per group. We also perform a Wilcoxon ranked sum test to establish significance in differences seen between groups.”

10. The authors mentioned that the MGPipe pipeline is reproducible. Can you provide a detailed operation tutorial or supplementary documentation to facilitate the use of other researchers? Has the compatibility of the pipeline in different sequencing platforms or cohorts of different sizes been tested?

This has been clarified in the Methods section: “The complete pipeline was implemented as a single shell wrapper script that automates each stage, manages R and Python dependencies via Conda, and maintains a unified logging structure for reproducibility. In this work, we applied MGPipe uniformly to all samples, including both sequencing batches by differential platform handling, ensuring standardized processing throughout ... To assess and correct for batch effects in the taxonomic abundance data due to differing Illumina platform sequencing between batches, we conducted a two-stage analysis combining permutational multivariate analysis of variance (PERMANOVA) with linear modeling”. The code repository houses a detailed operation tutorial in the README and is provided in the Data Availability statement through <https://github.com/ginnymortensen/MGPipe>. The pipeline was initially tested on the first batch of the data, and then used on the full cohort with both the first and second batches. The first and second batches were sequenced with two different Illumina platforms, and the pipeline accounts for this. The pipeline is described in the code repository to only be compatible with short, paired-end reads (NGS).

Re: Spectrum03155-25R1 (Metagenomic profiling and predictive modeling of the gut microbiome reveal signatures of gestational disease)

Dear Dr. David Haas:

Your manuscript has been accepted, and I am forwarding it to the ASM production staff for publication. Your paper will first be checked to make sure all elements meet the technical requirements. ASM staff will contact you if anything needs to be revised before copyediting and production can begin. Otherwise, you will be notified when your proofs are ready to be viewed.

Sincerely,
Ruixin Zhu
Editor
Microbiology Spectrum